# NYU CTF Bench: A Scalable Open-Source Benchmark Dataset for Evaluating Large Language Models in Offensive Security

## Motivation

**For what purpose was the dataset created? Was there a specific task in mind? Was there a specific gap that needed to be filled? Please provide a description.**

The dataset was created to evaluate the effectiveness of large language models (LLMs) in solving Capture the Flag (CTF) challenges within the domain of offensive security. There was a specific need to thoroughly assess the capabilities of LLMs in this context, as their potential for handling such tasks had not been systematically evaluated. The goal was to develop a scalable, open-source benchmark database specifically designed for these applications. This dataset includes diverse CTF challenges from popular competitions, with metadata to support LLM testing and adaptive learning.

The dataset addresses a critical gap by providing a comprehensive resource for the systematic evaluation of LLMs' performance in real-world cybersecurity tasks. It facilitates the comparison of LLM capabilities against human performance, offering insights into the potential of AI-driven solutions for real-world threat management. The development of this dataset and the accompanying automated framework allows for the continuous improvement and refinement of LLM-based approaches to vulnerability detection and resolution. By making the dataset open-source, the project aims to foster further research and development in this area, providing an ideal platform for developing, testing, and refining LLM-based approaches to cybersecurity challenges.

**Who created the dataset (e.g., which team, research group) and on behalf of which entity (e.g., company, institution, organization)?**

The dataset was created by astudents Minghao Shao, Sofija Jancheska, Meet Udeshi, Haoran Xi, Kimberly Milner, Boyuan Chen, and Max Yin, under the guidance of professors Brendan Dolan-Gavitt, Siddharth Garg, Prashanth Krishnamurthy, Farshad Khorrami, Ramesh Karri, and Muhammad Shafique from New York University(NYU) and New York University Abu Dhabi(NYUAD).

asThe CTF challenges were sourced from the annual Cybersecurity Awareness Week (CSAW) competition at NYU, created by various contributors each year. The students listed above compiled and validated these challenges from all previous global CSAW competitions by manually checking their setup and ensuring they remain solvable despite software changes. This work was conducted in collaboration with the OSIRIS Lab and the Center for Cybersecurity at NYU, which organize CSAW and attract global participation[1].

**Who funded the creation of the dataset? If there is an associated grant, please provide the name of the grantor and the grant name and number.**

The creation of the dataset was funded by the National Science Foundation (NSF) under grant number 2016650, the United Kingdom's Department for Science Innovation and Technology (DIST) under grant number G2-SCH-2024-02-13415, NYUAD Center for Cyber Security (CCS), funded by Tamkeen under the NYUAD Research Institute Award G1104, NYU Abu Dhabi Center for AI and Robotics CG010, Office of Naval Research N00014-22-1-2153 and ARO W911NF-22-1-0028.

**Any other comments?**
None.

## Composition

**What do the instances that comprise the dataset represent (e.g., documents, photos, people, countries)? Are there multiple types of instances (e.g., movies, users, and ratings; people and interactions between them; nodes and edges)? Please provide a description.**

The instances of the dataset represent CTF challenges from CSAW's CTF competitions hosted over a span of seven years.

Each CTF challenge consists of two required components: a JSON file and all relevant files provided by the challenge's author. The JSON file includes details about the challenge, such as its name, category, a brief description, and the flag, which is visible to our framework but not to the model. This file lists all necessary files required to solve the challenge, which may come in various formats, including images, code files, binary executables, or text files with hints about the challenge. In addition, for challenges requiring a Docker image to be deployed on the server side, we include an docker compose file in yml format. This image is loaded by our framework at the beginning of the challenge-solving process from docker compose and the images were pulled from the pre-built images on our Docker Hub. Some challenges are also coupled with a README file provided by the author, offering a textual description in plaintext or Markdown format. While this file aids in human understanding of the challenge, it is not utilized by the framework.

Finally, each CTF challenge belongs to its own category based on the type of skills and knowledge needed to solve it. The six categories are: cryptography, forensics, pwn, reverse engineering, web, and miscellaneous. There are no interactions between these categories; each challenge is self-contained and categorized independently.

**How many instances are there in total (of each type, if appropriate)?**
The total number of instances in the dataset is 200. The distribution of instances across each category is as follows: there are 52 challenges in the cryptography category, 15 in forensics, 39 in pwn, 51 in reverse engineering, 19 in web, and 24 in the miscellaneous category.

**Does the dataset contain all possible instances or is it a sample (not necessarily random) of instances from a larger set? If the dataset is a sample, then what is the larger set? Is the sample representative of the larger set (e.g., geographic coverage)? If so, please describe how this representativeness was validated/verified. If it is not representative of the larger set, please describe why not (e.g., to cover a more diverse range of instances, because instances were withheld or unavailable).**
Our initial dataset contained 568 CTF challenges sourced from CSAW's CTF competitions spanning 15 years. To create a more manageable and up-to-date dataset, we focused on the most recent seven years. From these, we successfully validated 200 challenges across six distinct categories.

This means that not all of the original 568 challenges can currently be solved by a human. Some reasons for this include outdated versions of the challenge requirements needed for the setup, missing challenge files, or unsolvable challenges. Thus, our current dataset comprises 200 challenges that are confirmed to be solvable in real-time by a human participant.

The dataset is not a random sample but a curated subset of the larger set, chosen for their solvability and relevance. These challenges were created manually and are expected to grow in number each year as we continue to gather challenges from other CTFs. The representativeness of this sample was ensured by selecting challenges from a variety of categories and difficulty levels, reflecting the diversity and scope of the original set.

**What data does each instance consist of? "Raw" data (e.g., unprocessed text or images) or features? In either case, provide a description.**
The data included in each instance may be presented in various formats as required by the challenge type. However, all challenges include a `.json` file as metadata, which contains detailed information about the challenge and the ground truth, which in this case is the flag.

For reverse engineering challenges, compiled binaries that need to be reversed are included, along with source files, if available, for reference by developers.

For pwn challenges, most instances contain a `docker-compose.yml` file to load pre-built Docker images from Docker Hub, which can be built on any Linux device. All relevant source files are included.

For cryptography, forensics, and miscellaneous challenges, the data type varies. Some instances may include a `docker-compose.yml` file to set up challenge containers, while others may contain necessary multimedia documents such as slides, images, or videos, depending on the specific instance.

For web challenges, all instances contain a `docker-compose.yml` file. Similar to reverse engineering instances, we include source code for developers' reference, though this source code is masked during evaluation when testing agent systems.

**Is there a label or target associated with each instance? If so, please provide a description.**
Yes, there is a label or target associated with each instance. The flags of the CTF challenges serve as the targets or labels. These flags are the "solutions" of a CTF, proving that the challenge has been successfully solved. All CTF challenges in our dataset contain the flag in their `challenge.json` file.

Each challenge belongs to one of six categories: cryptography, forensics, pwn, web, miscellaneous, and reverse engineering. Additionally, each challenge carries its "birth" information, including the year the challenge was created, whether it was used in the qualifying or final round of the CTF competition, and the score worth of each challenge for scoring purposes in the competition. These attributes serve as additional indirect labels, organizing the challenges and providing context for categorizing and evaluating them.

**Is any information missing from individual instances? If so, please provide a description, explaining why this information is missing (e.g., because it was unavailable). This does not include intentionally removed information, but might include, e.g., redacted text.**
There is no missing information in the individual instances. All necessary data required for solving the CTF challenges, including the flags, metadata, and associated files, have been thoroughly validated and included to ensure completeness and usability.

**Are relationships between individual instances made explicit (e.g., users' movie ratings, social network links)? Please describe how these relationships are made explicit.**
No relationships between individual instances are made explicit in the dataset. Each CTF challenge is a standalone instance, with no links or dependencies between challenges. This design ensures that each challenge can be evaluated independently, focusing on

the skills and knowledge required to solve it.

**Are there recommended data splits (e.g., training, development/validation, testing)? If so, please provide a description of these splits, explaining the rationale behind them.**
This dataset serves as a benchmark for evaluating the ability of LLMs in task planning with CTF challenges through automation. Consequently, there are no recommended data splits for training, development/validation, or testing for the benchmark dataset. The primary purpose of the dataset is to provide a consistent and comprehensive set of challenges to assess LLM performance in a standardized manner. But we also released a development dataset recently to help developers of large language model to enhance the capability of automated task planning and attacking under cybersecurity scenario for their models and agents.

**Are there any errors, sources of noise, or redundancies in the dataset? If so, please provide a description.**
All the challenges in the dataset have undergone thorough quality assurance processes. Consequently, the dataset is free from errors, sources of noise, and redundancies. Each challenge has been meticulously validated to ensure accuracy and consistency, maintaining the integrity of the dataset.

**Is the dataset self-contained, or does it link to or otherwise rely on external resources (e.g., websites, tweets, other datasets)? If it links to or relies on external resources, a) are there guarantees that they will exist, and remain constant, over time; b) are there official archival versions of the complete dataset (i.e., including the external resources as they existed at the time the dataset was created); c) are there any restrictions (e.g., licenses, fees) associated with any of the external resources that might apply to a dataset consumer? Please provide descriptions of all external resources and any restrictions associated with them, as well as links or other access points, as appropriate.**
The dataset is not entirely self-contained, as it relies on external resources hosted on Docker Hub for deploying all Docker containers used in this dataset.

a) Our Docker Hub namespace will remain constant and continues to exist over time, ensuring availability and consistency of Docker containers.

b) We have released our GitHub repository for public access to this dataset, which includes references to the necessary Docker containers. The GitHub repository can be accessed at LLM_CTF_Database.

c) There are no fees associated with accessing the external resources, and the dataset is distributed under the Apache License 2.0.

For further details and access to the external resources, please refer to our GitHub repository.

**Does the dataset contain data that might be considered confidential (e.g., data that is protected by legal privilege or by doctor–patient confidentiality, data that includes the content of individuals' non-public communications)? If so, please provide a description.**
No, the dataset does not contain any confidential data.

**Does the dataset contain data that, if viewed directly, might be offensive, insulting, threatening, or might otherwise cause anxiety? If so, please describe why.**
No, the dataset does not contain any data that might be offensive, insulting, threatening, or cause anxiety. All content within the dataset has been carefully curated to ensure it is appropriate and professional, focusing solely on technical challenges related to cybersecurity without any potentially harmful material.

**Does the dataset identify any subpopulations (e.g., by age, gender)? If so, please describe how these subpopulations are identified and provide a description of their respective distributions within the dataset.**
No, the dataset does not identify or target any specific subpopulations. It is designed to be universally applicable and does not include any demographic information such as age, gender, or other identifiers that could be used to categorize subpopulations. The focus remains solely on the technical aspects of CSAW's CTF competition.

**Is it possible to identify individuals (i.e., one or more natural persons), either directly or indirectly (i.e., in combination with other data) from the dataset? If so, please describe how.**
Yes, in the benchmark dataset, the README file lists the nicknames of the original authors for certain data instances, enabling users to identify the creators of each challenge for reference purposes. To ensure the privacy and anonymity of contributors, their real names are not disclosed. However, where available, the inclusion of the authors' nicknames allows for indirect recognition of their contributions.

**Does the dataset contain data that might be considered sensitive in any way (e.g., data that reveals race or ethnic origins, sexual orientations, religious beliefs, political opinions or union memberships, or locations; financial or health data; biometric or genetic data; forms of government identification, such as social security numbers; criminal history)? If so, please provide a description.**
No, the dataset does not contain any sensitive data. It strictly includes technical information related to CTF challenges and does not involve any personal, demographic, or sensitive information such as race or ethnic origins, sexual orientations, religious beliefs, political opinions, union memberships, locations, financial or health data, biometric or genetic data, government identification forms, or criminal history.

**Any other comments?**
None.

# Collection Process

**How was the data associated with each instance acquired? Was the data directly observable (e.g., raw text, movie ratings), reported by subjects (e.g., survey responses), or indirectly inferred/derived from other data (e.g., part-of-speech tags, model-based guesses for age or language)? If the data was reported by subjects or indirectly inferred/derived from other data, was the data validated/verified? If so, please describe how.**
Each instance in the dataset is organized into folders labeled with the year of creation, the event (CSAW Finals or CSAW Quals), the challenge category, and the challenge name. All metadata and README files are stored in their respective instance folders and are human-readable in raw text format. The dataset is a curated and quality-assured version derived from previous CSAW CTF competitions with standardized formatting. Validation and verification were conducted manually by following solvers included in the original challenge or by referencing write-ups to ensure these challenges are solvable by agent systems.

**What mechanisms or procedures were used to collect the data (e.g., hardware apparatuses or sensors, manual human curation, software programs, software APIs)? How were these mechanisms or procedures validated?**
The CTF challenges in the dataset were created by members of NYU's OSIRIS Lab and Center for Cybersecurity, as well as industry professionals. The dataset includes 200 validated challenges from the past seven years of CSAW CTF competitions. To ensure the challenges are solvable, students whose names were listed earlier repaired broken files, re-solved challenges using previous write-ups, and included solvers. This manual curation and validation process ensures the dataset is reliable for evaluating large language model-based AI agents in task planning or for other CTF competitions.

**If the dataset is a sample from a larger set, what was the sampling strategy (e.g., deterministic, probabilistic with specific sampling probabilities)?**
The dataset is a sample from a larger set originally comprising 568 CTF challenges sourced from CSAW's CTF competitions spanning 15 years. The sampling strategy involved validating and selecting challenges that are currently solvable by human participants. Out of the initial 568 challenges, 200 were successfully validated across six distinct categories. This selection process was necessary due to factors such as outdated challenge requirements, missing files, or un-

solvable challenges. Consequently, the current dataset contains 200 challenges confirmed to be solvable in real time by a human participant. These challenges were manually curated and are expected to increase in number as more challenges are gathered from future CSAW CTF and other CTF events.

**Who was involved in the data collection process (e.g., students, crowdworkers, contractors) and how were they compensated (e.g., how much were crowd workers paid)?**
The data collection process was conducted by students from NYU and NYU Abu Dhabi as part of their studies. These students were not compensated for their work, as it was part of their academic curriculum and research activities.

**Over what timeframe was the data collected? Does this timeframe match the creation timeframe of the data associated with the instances (e.g., recent crawl of old news articles)? If not, please describe the timeframe in which the data associated with the instances was created.**
The data was collected over a timeframe spanning the past 15 years, encompassing challenges from CSAW's CTF competitions. However, the dataset includes only the most recent seven years' worth of challenges that have been validated as currently solvable by human participants. This collection timeframe matches the creation timeframe of the data, as the dataset consists of CTF challenges created and used during these specific CSAW CTF events. The selection process involved repairing, re-solving, and validating these challenges to ensure they are up-to-date and solvable, aligning the dataset's collection period with the actual creation period of the challenges.

**Were any ethical review processes conducted (e.g., by an institutional review board)? If so, please provide a description of these review processes, including the outcomes, as well as a link or other access point to any supporting documentation.**
No formal ethical review processes were conducted for the creation of this dataset. However, we have reflected on the ethical considerations of using our dataset in the main paper.

**Did you collect the data from the individuals in question directly, or obtain it via third parties or other sources (e.g., websites)?**
No data from individuals was collected in the creation of the dataset.

**Were the individuals in question notified about the data collection? If so, please describe (or show with screenshots or other information) how notice was provided, and provide a link or other access point to, or otherwise reproduce, the exact language of the notification itself.**
N/A.

**Did the individuals in question consent to the**

collection and use of their data? If so, please describe (or show with screenshots or other information) how consent was requested and provided, and provide a link or other access point to, or otherwise reproduce, the exact language to which the individuals consented.

N/A.

**If consent was obtained, were the consenting individuals provided with a mechanism to revoke their consent in the future or for certain uses? If so, please provide a description, and a link or other access point to the mechanism.**

N/A.

**Has an analysis of the potential impact of the dataset and its use on data subjects (e.g., a data protection impact analysis) been conducted? If so, please provide a description of this analysis, including the outcomes, as well as a link to any supporting documentation.**

We conducted an initial evaluation of our dataset on 5 different LLMs using our proposed framework. The analysis involved utilizing these models within an automated workflow to assess their capabilities in solving CTF challenges through task planning. We measured their performance based on the percentage of challenges successfully solved. Additionally, we performed a basic analysis to identify the reasons for model failures, which highlighted the limitations of the models' capabilities.

Since the dataset does not include any personal data, a formal data protection impact analysis was not necessary. The evaluation focused solely on the technical performance and limitations of the models in a controlled environment.

**Any other comments?**

None.

## Preprocessing/cleaning/labeling

**Was any preprocessing/cleaning/labeling of the data done (e.g., discretization or bucketing, tokenization, part-of-speech tagging, SIFT feature extraction, removal of instances, processing of missing values)? If so, please provide a description. If not, you may skip the remaining questions in this section.**

We pre-built all Docker containers needed for the CTF challenge servers in advance and deployed them on our Docker Hub. For each challenge with a Dockerfile, there is a docker-compose file containing all the necessary configurations to pull these images. The real flags, serving as the ground truth for these challenges, are labeled and included in the metadata of each instance in a `challenge.json` file.

**Was the "raw" data saved in addition to the preprocessed/cleaned/labeled data (e.g., to support unanticipated future uses)? If so, please provide a link or other access point to the "raw" data.**

All raw data, including the source code of binary programs, README files, and original Dockerfiles, are included with each instance along with the dataset files. The challenge.json metadata file lists all the necessary files, ensuring the agent system can load them based on the metadata. This setup preserves both the raw and preprocessed data for each CTF challenge.

**Is the software that was used to preprocess/clean/label the data available? If so, please provide a link or other access point.**

Yes, the software used to preprocess, clean, and label the data is available. We have released our complete automation framework, which includes the dataset loader and the code for conducting the evaluation experiments, all of which are fully explained in our main paper.

**Any other comments?**

None.

## Uses

**Has the dataset been used for any tasks already? If so, please provide a description.**

Yes, the dataset has already been used for specific tasks. We conducted a baseline evaluation using our automation framework to assess five LLMs' capabilities in solving CTF challenges, measuring performance by the percentage of challenges solved and analyzing model failures to highlight limitations. Additionally, the paper An Empirical Evaluation of LLMs for Solving Offensive Security Challenges utilizes a partition of this dataset, specifically CTF challenges from the 2023 Quals, for its evaluation experiments and competition.

**Is there a repository that links to any or all papers or systems that use the dataset? If so, please provide a link or other access point.**

Yes, there is a repository that links to papers and systems using the dataset. The database repository can be accessed at LLM_CTF_Database. Additionally, the starter framework repository is available at llm_ctf_automation.

**What (other) tasks could dataset be used for?**

The dataset can be used for any type of automation and task planning for CTF challenges. Additionally, it can serve educational purposes by providing a comprehensive set of challenges for students and professionals to practice and enhance their cybersecurity skills.

**Is there anything about the composition of the dataset or the way it was collected and preprocessed/cleaned/labeled that might impact future uses? For example, is there anything that a dataset consumer might need to know to avoid uses that could result in unfair treat-**

ment of individuals or groups (e.g., stereotyping, quality of service issues) or other risks or harms (e.g., legal risks, financial harms)? If so, please provide a description. Is there anything a dataset consumer could do to mitigate these risks or harms?

The future versions of the dataset will follow the same paradigm as our current version, ensuring consistency when new instances are added. It is important for users to avoid using this dataset's content, including all solvers and solutions, to attack real-world systems. This dataset is intended for educational and research purposes, such as improving cybersecurity skills and developing automated systems for solving CTF challenges, not for malicious activities. To mitigate risks, users should adhere to ethical guidelines and use the dataset responsibly within controlled environments, keeping in mind the ethical reflections discussed in our main paper.

**Are there tasks for which the dataset should not be used? If so, please provide a description.**

The dataset should not be used for certain tasks. Specifically, it is prohibited to use the dataset for training models aimed at attacking real-world computer systems, as well as employing the transcripts generated by the framework for such purposes. This restriction ensures that the dataset and associated transcripts are not misapplied in harmful contexts where they might lack the necessary robustness or accuracy for effective cybersecurity measures. Instead, the dataset is intended for evaluation and benchmarking purposes within controlled environments to assess the capabilities of LLMs in solving offensive security challenges. While it can be used for training, care must be taken to avoid any misuse in harmful applications.

**Any other comments?**
None.

# Distribution

**Will the dataset be distributed to third parties outside of the entity (e.g., company, institution, organization) on behalf of which the dataset was created? If so, provide a description.**

Yes, the dataset is publicly accessible, allowing all parties to obtain the source code of the dataset along with our starter framework via GitHub. The repositories are distributed under the Apache 2.0 license, so users must comply with its terms.

**How will the dataset will be distributed (e.g., tarball on website, API, GitHub)? Does the dataset have a digital object identifier (DOI)?**

The dataset is published on GitHub at `https://github.com/NYU-LLM-CTF/LLM_CTF_Database`. Currently, it does not have a digital object identifier (DOI).

**When will the dataset be distributed?**

The dataset is already available and publicly accessible on GitHub. All Docker images can be accessed via the `docker-compose.yml` file included in the source code of instances, using the corresponding namespace "llmctf" of our Docker Hub account along with the challenge name.

**Will the dataset be distributed under a copyright or other intellectual property (IP) license, and/or under applicable terms of use (ToU)? If so, please describe this license and/or ToU, and provide a link or other access point to, or otherwise reproduce, any relevant licensing terms or ToU, as well as any fees associated with these restrictions.**

Yes, the dataset is distributed under the Apache 2.0 license. There are no fees associated with these restrictions. You can access the license terms at `https://www.apache.org/licenses/LICENSE-2.0`.

**Have any third parties imposed IP-based or other restrictions on the data associated with the instances? If so, please describe these restrictions, and provide a link or other access point to, or otherwise reproduce, any relevant licensing terms, as well as any fees associated with these restrictions.**

No, there are no IP-based or other restrictions imposed by third parties on the data associated with the instances. As long as third parties have access to Docker Hub and GitHub, the dataset source and images will be accessible.

**Do any export controls or other regulatory restrictions apply to the dataset or to individual instances? If so, please describe these restrictions, and provide a link or other access point to, or otherwise reproduce, any supporting documentation.**

No, there are no export controls or other regulatory restrictions that apply to the dataset or to individual instances.

**Any other comments?**
None.

# Maintenance

**Who will be supporting/hosting/maintaining the dataset?**

NYU's Center for Cybersecurity, in collaboration with NYU's OSIRIS Lab, will be supporting and maintaining the dataset. This effort will be supervised by Ramesh Karri, the director of the Center for Cybersecurity.

**How can the owner/curator/manager of the dataset be contacted (e.g., email address)?**

The manager of the dataset can be contacted via email at rkarri@nyu.edu.

**Is there an erratum? If so, please provide a link or other access point.**
No, there is no erratum available.

**Will the dataset be updated (e.g., to correct labeling errors, add new instances, delete instances)? If so, please describe how often, by whom, and how updates will be communicated to dataset consumers ?**
The dataset will be updated annually, in alignment with CSAW's CTF competition from which the challenges were sourced. Each year, typically in November, new CTF challenges from both the qualifying and final rounds of the competition will be added to the dataset. Additionally, contributions from the public are welcome, following the established data structure and organization. Furthermore, our larger dataset contains CTF challenges that have not yet been published as they are still being validated. As these challenges are validated and confirmed solvable, they will be added to the dataset using the current format.

**If the dataset relates to people, are there applicable limits on the retention of the data associated with the instances (e.g., were the individuals in question told that their data would be retained for a fixed period of time and then deleted)? If so, please describe these limits and explain how they will be enforced.**
N/A.

**Will older versions of the dataset continue to be supported/hosted/maintained? If so, please describe how. If not, please describe how its obsolescence will be communicated to dataset consumers.**
We deliver the dataset source code via GitHub, where all commits reflect changes to the dataset versions. We use Docker Hub to release our pre-built Docker images, which will be maintained and updated regularly. However, older versions will be overwritten on Docker Hub, and previous versions of the images may not be available if there is an update.

**If others want to extend/augment/build on/contribute to the dataset, is there a mechanism for them to do so? If so, please provide a description. Will these contributions be validated/verified? If so, please describe how. If not, why not? Is there a process for communicating/distributing these contributions to dataset consumers? If so, please provide a description.**
Yes, others can easily extend or augment the dataset by adding more challenges. To do so, contributors need to create new challenge folders, with each challenge placed in its own folder, following our existing structure. For each challenge, contributors must create a metadata file named `challenge.json`, which includes the following compulsory fields: `name` (the name of the challenge), `files` (a list of files neces-sary to solve the CTF), `description` (a brief description of the challenge content), `flag` (the ground truth, which is the real label of the challenge), and `category` (the category of the challenge). Contributors can add categories, as long as the structure of the metadata follows our format. For challenges that include a Docker container as a server, developers should create a `docker-compose.yml` file and include the `image` field to pull the image from Docker Hub. The namespace of the Docker Hub can vary, as long as the prebuilt Docker images can be successfully pulled. All contributions will be validated and verified to ensure they adhere to the required structure and are solvable. Once validated, these contributions will be communicated and distributed to dataset consumers through updates in our GitHub repository. This ensures that the dataset remains consistent and reliable while allowing for community-driven expansion.

**Any other comments?**
None.

# Impact and Challenges

The dataset has a significant impact on the evaluation of LLMs in offensive security. By compiling 200 validated CTF challenges across six categories from a pool of 567 challenges over seven years, this dataset provides a comprehensive and diverse set of problems for testing the capabilities of LLMs. This diverse range of challenges allows for a thorough assessment of LLMs' strengths and weaknesses in various aspects of cybersecurity, fostering a deeper understanding of how these models can be applied effectively. The dataset's scalability and inclusion of real-world challenges make it an invaluable resource for researchers and practitioners aiming to enhance the robustness and reliability of LLMs in cybersecurity applications and task planning.

One of the main challenges of the dataset is ensuring its continued growth and balance. Expanding the dataset while maintaining a representative distribution across different categories is crucial for comprehensive evaluations. Additionally, validating new challenges to ensure they are solvable and relevant remains a complex and time-consuming task. Another challenge is the integration of new cybersecurity tools and techniques to keep the dataset up-to-date with the evolving landscape of cybersecurity threats. Addressing these challenges is essential to maintain the dataset's relevance and effectiveness in training and evaluating LLMs for real-world offensive security tasks.