# OpenReview forum: "NYU CTF Bench: A Scalable Open-Source Benchmark Dataset for Evaluating LLMs in Offensive Security"
_NeurIPS.cc/2024/Datasets_and_Benchmarks_Track — NeurIPS 2024 Track Datasets and Benchmarks Poster_

### Official Review · Reviewer_RHcs · 2024-07-22
**Review for NYU CTF Dataset: A Scalable Open-Source Benchmark Dataset for Evaluating LLMs in Offensive Security**

**Rating:** 8
**Confidence:** 4

**Review:**

The paper introduces a robust CTF dataset with six distinct categories of challenges, each requiring different skills and tools. This variety allows for a comprehensive assessment of LLM capabilities in offensive security. The dataset and framework are well-documented, clearly described, and visualized, ensuring ease of understanding and use.

The dataset's construction is sound, utilizing Docker images for necessary environments and JSON files for metadata. The automatic evaluation framework is logically structured and clearly explained, supporting reproducibility.

Opportunities for improvement include incorporating more LLMs for broader skill comparison and adding various CTF-like challenges or standardizing user submissions to enhance dataset diversity.

The paper is well-written, clearly structured, and succinct. It stands out for its unique automatic framework for LLM evaluation and diverse challenges requiring tool usage. Ethical considerations are thoroughly addressed, highlighting risks of LLM misuse and advocating for explainable AI and ethical education in cybersecurity.

The relationship between Figure 3 and the textual illustrations could be more clear. The image contains many details that are hard to map with the text. For example, the text mentions five modules, which cannot be identified in the figure. An extended comparison with related work would be beneficial. Additionally, a more in-depth evaluation of the LLMs' failure conditions regarding alignment filters and an explanation for the different performances in Table 5 could be addressed.

Minor issue:

- At the end of page 8, the table is referenced as 8.1, which should be 8.

**Strengths:**

- Both the dataset as well as the framework are clearly described and visualized, making it easy to understand its exact components.
- Variety of different categories to allow for broader generalizability of LLMs CTF solving performance enabling a good possibility of assessing the capabilities of LLMs.
- Framework to evaluate different models allowing for using the dataset with less effort
- Initial evaluation with a comparison of the performance of different LLMs including failure points and a comparison with human CTF solvers

**Additional Feedback:**

No additional feedback

**Clarity:**

The overall structure is well done and the writing quality is adequate. The paper is well-written, clearly structured, and formulated to the point. However, the relationship between Figure 3 and the textual explanation should be improved. It would help if in Figure 3 the categories align with the paragraphs.

**Correctness:**

The claims made in the submission are correct. The dataset is constructed in a sound way and in a meaningful manner. Docker images are used to encapsulate necessary environments for each single challenge, while JSON files are used to provide metadata information for support files and tools. The evaluation is performed correctly and the experiment design is appropriate. The automatic evaluation framework is described clearly and soundly.

**Documentation:**

The dataset consists only of challenges from their own New York University CTF competitions. The authors also attach a complete list of all challenges in the dataset as well as the included software in their framework. An URL for reviewers is included and contains both the dataset as well as the evaluation framework. The detail on data collection and organization is sufficient. The submission includes documentation to support reproducibility.

**Ethics:**

The authors emphasize the ethical considerations of using CTF challenges to benchmark LLMs in cybersecurity. While LLMs have advanced in providing accurate responses, risks of misuse for social engineering and malware creation remain. The legal framework struggles to keep up with AI developments, and researchers advocate for explainable AI to ensure transparency and accountability. Integrating LLMs in CTF challenges requires ethical education to bridge the gap between cybersecurity training and AI advancements. Understanding these ethical implications is crucial. The authors also emphasize the exploitation for social engineering or malware creation, revealing the dual nature of AI as both a tool and a potential threat.

**Limitations:**

The paper mostly addresses the limitations of their work adequately, however, I would emphasize more how the objective is in direct conflict with the ethical alignment of many LLMs such as GPT-4, since this task can easily be misused to convert a CTF game into a real attack.  However, I would argue it is still reasonable to conduct such research for academic purposes and suggest some measures to minimize harm. Additionally, the authors discuss both their ethical concerns as well as current limitations in their dataset. They emphasize that understanding the ethical implications of deploying LLMs in cybersecurity tasks is essential for decision-makers and that their dataset shows some imbalances for certain challenges.

**Opportunities For Improvement:**

Limitations explained by the authors:

- Limited categories such as incidence response missing
- Imbalanced number of samples per category
- Limited tool use support
- Limited number of LLMs are supported

Additional limitations:

- No evaluation of the impact of the model's alignment, for example via fine- or instruction tuning, on the performance
- No explanations as to why the models performed so differently for CSAW 2022 vs 2023 and Qual vs Final. Can memorization be the issue?

**Relation To Prior Work:**

This work differs mainly from its unique aspects like an automatic framework to evaluate LLMs on the said dataset, as well as very unique and plenty of challenges requiring tool usage. Comparing the approach to existing work in a table with listed criteria is a good way to emphasize your contribution in a structured manner. However, I would include recent peer-reviewed work such as [1] and [2]and outline the advantages of your work.

[1] Deng et al. “PentestGPT: Evaluating and Harnessing Large Language Models for Automated Penetration Testing", USENIX Security Symposium 2024

[2] Dhananjai et al. “Automated Penetration Testing using Large Language Models", International Journal of Science and Research (IJSR) 2024

**Summary And Contributions:**

The paper presents a method for assessing the capabilities of Large Language Models (LLMs) in solving cybersecurity Capture the Flag (CTF) challenges by developing an open-source benchmark database. This database focuses on scalability and includes metadata for LLM testing and adaptive learning, compiling a diverse range of CTF challenges from various competitions. It integrates advanced function-calling capabilities of LLMs to build a fully automated system. The paper evaluates the performance of five different LLMs, including both black-box and open-source models, in solving these challenges, providing insights into the potential of LLMs for AI-driven cybersecurity solutions.

The authors contribute by creating an open dataset with various CTF challenges and a framework to test LLMs on their dataset. The toolkit enhances LLM capabilities for solutions using tool-calling functionalities. The dataset consists of challenges from different categories such as reverse engineering and forensics. The authors also provide a framework to automatically evaluate several existing models on their dataset, including both commercial and open-source LLMs. They perform an initial evaluation with several models to compare their CTF solving abilities. Both the dataset and the framework are publicly available.

---

> ### Author Rebuttal · Authors · 2024-08-16
>
> Dear reviewer RHcs,
>
> We value your feedback on our paper and have revised it accordingly. Below, we offer detailed responses to each comment provided. Please note that comments are out of order as similar comments are clubbed together, followed by a common response.
>
> > Minor issue:
> > - At the end of page 8, the table is referenced as 8.1, which should be 8.
>
> We have fixed the table reference on page 8 in the revised paper.
>
> > The relationship between Figure 3 and the textual illustrations could be more clear. The image contains many details that are hard to map with the text. For example, the text mentions five modules, which cannot be identified in the figure.
>
> > However, the relationship between Figure 3 and the textual explanation should be improved. It would help if in Figure 3 the categories align with the paragraphs.
>
> In the revised Figure 3, we have added numbers 1 to 5 corresponding with the 5 modules in Section 3 to clearly map the relationship between the figure and the text.
>
> > No evaluation of the impact of the model's alignment, for example via fine- or instruction tuning, on the performance
>
> The scope of this paper is to present the NYU CTF dataset for evaluation of LLM agents on cybersecurity challenges. We have provided a preliminary evaluation of current LLMs using our automation framework, which can be considered as a baseline. Further evaluations of specific properties such as alignment, and impact of techniques like fine-tuning, are avenues for future work that can use our dataset and compare with our baseline results.
>
> > Additionally, a more in-depth evaluation of the LLMs' failure conditions regarding alignment filters and an explanation for the different performances in Table 5 could be addressed.
>
> > No explanations as to why the models performed so differently for CSAW 2022 vs 2023 and Qual vs Final. Can memorization be the issue?
>
> From our analysis of the transcripts of success and failure cases, we see the following reasons for differences in accuracy:
>
> 1. Task complexity: CTF challenges are open-ended and require multiple rounds of vulnerability analysis and exploitation. For LLMs with different capabilities (e.g., context length, model size), this leads to different approaches to solve the CTF and produces different accuracies.
> 2. Challenge difficulty: In addition to complexity of CTFs in general, each challenge differs in difficulty as indicated by its point score. Figure 1 shows that Finals challenges are more difficult than the Quals challenges of each year. The point score is assigned to each challenge by the competition organizers.
> 3. Randomness in LLM evaluation: We use the default temperature settings of each LLM, which leads to differences in outputs with the same challenge. Due to the feedback loop and autonomous behavior, we see divergence of the approaches taken by each LLM across multiple runs.
>
> We have modified the discussion in Section 4.1 with these points.
>
> > I would emphasize more how the objective is in direct conflict with the ethical alignment of many LLMs such as GPT-4, since this task can easily be misused to convert a CTF game into a real attack. However, I would argue it is still reasonable to conduct such research for academic purposes and suggest some measures to minimize harm.
>
> We agree with the reviewer regarding the ethical conflict in CTFs. However, the benefit of CTF competitions in cybersecurity education is well-accepted in the security community. In our preliminary evaluation, we do not observe any instance where the LLM refuses to solve a CTF challenge due to ethical conflicts, which indicates that current LLMs understand the educational context of CTFs. While this behavior offers potential for misuse, future research in LLM alignment and safety can focus on detecting simulated versus real attack scenarios to further safeguard against LLM misuse. We have added these points to Section 4.2 in the revised paper.
>
>
> > An extended comparison with related work would be beneficial.
>
> > This work differs mainly from its unique aspects like an automatic framework to evaluate LLMs on the said dataset, as well as very unique and plenty of challenges requiring tool usage. Comparing the approach to existing work in a table with listed criteria is a good way to emphasize your contribution in a structured manner. However, I would include recent peer-reviewed work such as [1] and [2]and outline the advantages of your work.
>
> We have modified the related work section in the revised paper to highlight the differences of our dataset with those used in related papers on LLM agents for cybersecurity. We have included recently published work like PentestGPT in our comparison.

---

> > ### Comment · Reviewer_RHcs · 2024-08-19
> >
> > Thank you for the clarifications. I still fully support this work

---

### Official Review · Reviewer_82Xy · 2024-07-23
**dataset consisting of Capture the flag exercises around cybersecurity use-cases**

**Rating:** 6
**Confidence:** 5
**Correctness:** Yes.
**Clarity:** Yes.

**Review:**

the dataset is well defined and even though the challenges are only 200, the good part is that each challenge has its own docker file which allows for easy setup of the infrastructure and play it out. Especially for models, this will be beneficial. There are a few more Github resources consisting of CTF challenges and solutions but they are not well maintained.
The challenge here is a more continuous maintenance of the challenges, its docker images and then feeding the information into the models for bench marking.

**Strengths:**

Good work getting working docker images for each of the 200 challenges. Without a viable means for spinning up the challenge and trying to solve it out, benchmarking would be difficult.

**Additional Feedback:**

NA

**Documentation:**

Yes. Github repo provided.

**Ethics:**

yes.

**Limitations:**

2. Dockerfile is a great start but requires continuous maintenance. This is also one of the points noted in the dataset collection that some of the older challenges were not included due to the fact that tracing back the tech stack wont be easy.

**Opportunities For Improvement:**

1. Instruction sets on how an autonomous system should approach at solving the challenges listed in the dataset.
2. A more targeted approach can be to start with a fixed branch(for eg. Forensics or Reverse engineering etc) and build a dataset around it instead of breaking it down into years. Then expand it to include more branches of CTF, expanding it into a branch distributed set than year distributed.

**Relation To Prior Work:**

This was a dataset so there was little reference to prior work.
They did refer to the prior work of the university in collecting and maintaining the CTF challenges.

**Summary And Contributions:**

The dataset comprises of 200 CTF challenges hosted by the university in the span of last 7 years. the dataset contains the docker images to build up the exercise and then JSON files for each challenge consisting of a flag which once found in the CTF, shows the successful completion of the CTF.

---

> ### Author Rebuttal · Authors · 2024-08-16
>
> Dear reviewer 82Xy,
>
> We value your feedback on our paper and have revised it accordingly. Below, we offer detailed responses to each comment provided.
>
> ## Opportunities for Improvement:
>
> > Instruction sets on how an autonomous system should approach at solving the challenges listed in the dataset.
>
> We believe you are referring to instructions to develop an autonomous system that can solve CTF challenges (if you are referring to a different aspect, please let us know). We have described our framework in Section 3, which can be referred to as a baseline for building more advanced autonomous systems. The implementation of tool usage functionalities, feedback loops, and prompt templates used in our framework are found in the framework repository (https://github.com/NYU-LLM-CTF/llm_ctf_automation). Additionally, summaries of transcripts of our framework’s solution of some challenges are given in Appendix A, which can aid future development.
>
> > A more targeted approach can be to start with a fixed branch(for eg. Forensics or Reverse engineering etc) and build a dataset around it instead of breaking it down into years. Then expand it to include more branches of CTF, expanding it into a branch distributed set than year distributed.
>
> The users of our dataset are free to pick any bifurcation that they prefer based on the agent they are developing or evaluating. The bifurcations for year and category have been provided in the dataset only as guidelines. Specialized LLM agents that are tailored to a specific task (such as web-based exploits) may pick a specific category (such as web) for evaluation. Others wanting to perform a smaller evaluation may pick a specific year to get challenges across categories from a single competition. Our dataset offers this versatility which makes it stand out over other CTF datasets tailored to specific categories.
>
> ## Limitations:
>
> > Dockerfile is a great start but requires continuous maintenance. This is also one of the points noted in the dataset collection that some of the older challenges were not included due to the fact that tracing back the tech stack wont be easy.
>
> To reduce the burden of maintenance, we have built and tested the Docker images of each of the 200 challenges and uploaded them on Docker hub (https://hub.docker.com/u/llmctf). These pre-built images are directly pulled and deployed when the framework loads a challenge, so there is no need to build the Docker images locally. This allows all challenges to be easily deployed in the future without having to maintain the Dockerfiles.

---

> > ### Author Response · Authors · 2024-08-31
> >
> > Dear reviewer 82Xy,
> >
> > A gentle reminder that we have updated the paper with extra references covered in the related work (section 1.3) and an extra appendix about comparing human and LLM approaches for a CTF challenge in Appendix C, the updated Arxiv link is attached: [https://arxiv.org/pdf/2406.05590](https://arxiv.org/pdf/2406.05590).

---

### Official Review · Reviewer_rkZ5 · 2024-07-25
**Interesting contribution to assess the ability of LLMs to solve CTF challenges**

**Rating:** 6
**Confidence:** 3

**Review:**

The dataset and associated framework provide a valuable platform for researchers to explore the potential of LLMs in addressing cybersecurity challenges. This has significant implications for both cybersecurity education and research, by fostering research into utilizing LLMs for vulnerability detection and exploitation in real-world scenarios, and to promote cybersecurity education.

However, in its current form, it is unclear how this dataset is a good approach to foster the research in this area, and develop benchmark to assess the capacity of LLMs to solve CTF. Its reliance on a complex infrastructure involves capacities of reasoning, planning and control that go beyond the CTF challenges. The fact that most of the data is likely to be already incorporated in the public dataset used to train LLMs also prevent a good understanding of what is actually assessed when the dataset is run.

This submission is nonetheless worthwhile and has the potential to be relevant to improve the use of LLMs for cybersecurity applications, provided changes are made to strengthen the motivations and the analysis of the dataset's capabilities.

**Strengths:**

- **Clear and well-organised dataset**: The dataset is well-organised. It is publicly accessible on Github, and straightforward to understand thanks to its clear structure. This ensures clarity and usability for the broader research community.
- **Accompanying framework**: The development of a framework that facilitates interaction with a working environment adds substantial value to the dataset. This practical tool allows researchers to directly test and evaluate LLM-based solutions within a relatively realistic cybersecurity context. This is all the more relevant as the dataset is relatively complex (scripts, docker files, data, etc.) and requires some knowledge to be properly used.
- **Impact on cybersecurity research and education**: The contribution addresses a need within the field by providing a comprehensive collection of CTF challenges covering diverse cybersecurity topics. The potential impact for the security community is significant, facilitating education but also opening avenues for research into vulnerability detection and exploitation using LLMs.

**Additional Feedback:**

The dataset explores the use of LLMs to solve CTF challenges in an automated way. I think it would be interesting to explore the use of LLMs to assist participants, and understand how the dataset could be restructured to assess the ability of LLMs to provide valuable insights in a problem and complement human reasoning.

UPDATE:
The responses made by the authors during the rebuttal phase provided clarifications to some of my concerns. In my view, the changes made in the revised version are not fully satisfactory (e.g., the related works section is still weak), but I think this work is of sufficient quality and impactful enough to be helpful for the research community, and therefore I recommend its acceptance.

**Clarity:**

The paper presents its findings in a manner that is both clear and concise, making it accessible to a broad audience of machine learning and cybersecurity experts.

Minor details:
- Univeristy -> University
- Missing space between Table/Figure and numbers
- It looks like samples in Table 3 are not present in the tables in Appendix.
- Many typos in the datasheet

**Correctness:**

The dataset is constructed in a sound way. It is not clear however how it can be used to develop a sound benchmark for LLMs, as it has been designed for human participants and may not be adapted to assess LLMs.

**Documentation:**

The documentation is clear and the datasheet provided alongside the submission gives many details about the content of the dataset, and how it was build. An unclear aspect concerns the ownership of copyright for the challenges, that have different authors, and whether the authors agreed to be included in the dataset.

The documentation regarding the automated framework that accompanies the dataset, and which is crucial to exploit it, is however missing. In particular, the maintenance of the tools over the years is a crucial aspect to ensure that the tools will remain compatible and usable by research groups.

**Ethics:**

The Ethics section is very broad and general, and while I do not see any significant concerns as most data are already public, it seems that  issues such as the potential use by malicious actors are not clearly stated, or the security risks for the non-expert user who would run the scripts present in the dataset.

**Limitations:**

The limitations of the dataset are not clearly stated, in particular with regards to the points mentioned above.

**Opportunities For Improvement:**

- **Potential data leakage bias**: Given that many CTF solutions are readily available online, including detailed write-ups by participants outlining the steps taken, there is a high probability that LLMs will have encountered these solutions during their training process, or will have access to that through search capabilities. This undermines the relevance of the dataset to properly assess the capacity to conduct cybersecurity tasks.

- **Clarification on scope and application**: from their initial experiments, it is not clear whether the dataset tests a  LLM's capability to solve CTF challenges, or its ability to execute commands within a working environment and processing outputs.

- **Dataset availability**: The dataset is already available on Github in other repositories (presumably from the same authors), but with the exact same format and content. It seems then that this version does not provide any novelty compared to what is already available and may only be the compilation of data used in the context of the competition.

**Relation To Prior Work:**

The "related works" section appears superficial, merely listing CTF competitions without explaining how they differ from the one that has been used to build the dataset. The related works section should also devote more attention to prior research involving LLMs in cybersecurity contexts, and provide concrete details about the methods employed in prior studies to evaluate LLMs on CTF challenges, and how their dataset will help fostering research in this area, highlighting  its unique aspects.

**Summary And Contributions:**

This paper makes several contributions to the field of cybersecurity education and research. It presents a publicly accessible dataset comprising 200 Capture-The-Flag (CTF) challenges spanning a wide array of cybersecurity domains, drawing upon past
challenges from the CSAW CTF competition. Additionally, the authors develop a framework enabling the execution of both open-source and black-box Large Language Models (LLMs) to tackle these CTF challenges, with the support of for external tool calls. A quick experimental section of the performance of mainstream LLMs on the dataset is provided, showing the current limited capabilities of models to solve the tasks.

---

> ### Author Rebuttal · Authors · 2024-08-16
>
> Dear Reviewer rkZ5,
>
> We value your feedback on our paper and have revised it accordingly. Below, we offer detailed responses to each comment provided.
> ## Opportunities for Improvement
> > Potential data leakage bias:…
>
> In our search for publicly available write ups for the 200 challenges we found partial and incomplete write ups (some with  inaccurate solutions) for only 80 of them. We found no write ups for the remaining 120 challenges. We will update the dataset with links to known write ups for these 80 problems to help in further research on memorization impact on LLM evaluations. Our results show that. GPT-3.5 achieves 1.25% accuracy on the 80 challenges with write-ups and 5.833% accuracy on the 120 challenges without write-ups, suggesting that the public write-ups did not impact the accuracy. Additionally, CTF challenges are open-ended with no standardized solutions; therefore, participants may approach the same challenges with different solution approaches. We have added a case study to the paper for a challenge where we compare the LLM agent’s solution with publicly available write-ups. We observe that the LLM agent uses a completely different approach to solve the challenge successfully. This case study, along with the accuracy results, indicates that the potential for data leakage to impact the accuracy of LLM agents on our dataset is low
> > Clarification on scope and application:…
>
> Our dataset evaluates the ability of LLM agents to solve CTF challenges by successfully capturing the correct flag, a task that requires a diverse set of analysis and execution skills. Merely executing commands is insufficient for high performance; agents must also perform high-level cybersecurity analysis, plan strategically, and execute exploits effectively. Our automation framework assesses agents based on their success in solving the challenge, not just on command execution. As outlined in the abstract and Section 4, our benchmark dataset enables the evaluation of LLM agents based on these criteria of success or failure.
> > Dataset availability:…
>
> The CSAW CTF challenges on GitHub included necessary files and Docker configurations but needed further additions and standardizations for usability. Here are the key novelties of our dataset:
> 1. Docker configuration: the challenges developed over multiple years, had outdated Docker configurations and network setups with overlapping ports and inaccessible domains, making the original Docker files unusable. We updated each challenge with current packages, standardized network configurations, built Docker images, and tested them for deployment. The images were then uploaded to Docker Hub, allowing the automation framework to directly pull these pre-built images instead of rebuilding them.
> 2. Standardization: We curated metadata for each of the 200 challenges, including a consistent description, flag details, and network settings. This metadata is crucial for using the challenges in an automation framework and verifying flag correctness. The standardized data structure is illustrated in Figure 2 of the paper.
> 3. Baseline automation framework: We developed a framework to load and deploy CTF challenges within a container environment, enabling LLM agents to operate and solve tasks. This framework supports researchers in creating more LLM agent systems and models, as detailed in Section 3 of our paper.
> ## Correctness
> > It is not clear however how it can be used to develop a sound benchmark for LLMs…
>
> CTF challenges simulate real-world attack scenarios to assess the ability to identify and exploit vulnerabilities, applicable to both humans and automated agents. This relevance is highlighted by initiatives like the DARPA Cyber Grand Challenge and DARPA AIxCC competitions. A benchmark dataset specifically formatted for LLM agents enables testing of their cybersecurity skills. CTFs involve complex tasks such as command line operations, environment setups, tool usage, and exploit implementation, requiring advanced planning and reasoning from the model. As discussed in Section 1.3, the impact of CTFs on cybersecurity education is widely recognized in the field. Testing LLM agents against challenges faced by humans allows us to compare their cybersecurity skills with human experts and other autonomous systems. Also, it would be contrary to the purpose of benchmarking LLM capabilities to test on challenges designed specifically for LLMs and not generally applicable to real-world scenarios.
> ## Clarity
> > Univeristy -> University…
>
> We have updated them in the revised paper.
> > It looks like samples in Table 3 are not present in the tables in Appendix.
>
> We included all the challenges in the appendix, since Table 3 is an example to show features and characteristics of each category covered, challenge names were not provided. The challenge names in Table 3 are: crypto: polly-crack-this 2022f; forensics: 1black0white 2023q; pwn – puffin (2023q); reverse: rebug 1 2023q; web: smug-dino 2023q; misc: AndroidDropper 2023q. (q: Quals, f: Finals)
> ## Relation to prior work
> > The "related works" section appears superficial…
>
> We revised the Related Work section to specify that the CTF frameworks discussed are intended for human competitions, while our framework deploys challenges for LLM agents. We’ve also referenced recent papers on LLM-based cybersecurity methods. Our dataset, encompassing six categories, is the most extensive collection of CTF challenges for LLM agents available. In contrast, other related studies concentrate on specific categories and offer fewer challenges for evaluation.
> > An unclear aspect concerns the ownership of copyright for the challenges…
>
> In our dataset, each challenge’s metadata contains the name and Github ID of the challenge authors to clearly attribute authorship. We sourced the challenges from the CSAW competition archives, which organizers have publicly shared on Github. Citations to the relevant Github repositories are included in the revised paper.

---

> > ### Comment · Reviewer_rkZ5 · 2024-08-19
> >
> > I would like to thank the authors for their detailed answers. Unless I'm mistaken, the revised version is not available, therefore I cannot assess how the changes have been implemented.
> >
> > Generally speaking, I would be convinced by the responses made by the authors. I would still have some concerns regarding the scope of the contribution, as in my opinion it fails as a benchmark to give a relevant score assessing the cybersecurity skills of LLMs. However, I acknowledge that the contribution is first and foremost a dataset and in that regards, it has its relevance and may be useful for research communities.
> >
> > Therefore, I would be willing to give a higher score and recommend the acceptance of the submission. I'm open for a review of the revised version if it is available.

---

> > > ### Author Rebuttal · Authors · 2024-08-22
> > >
> > > Dear reviewer rkZ5,
> > >
> > > We have uploaded the revised paper on Arxiv. Plese access it here [https://arxiv.org/pdf/2406.05590](https://arxiv.org/pdf/2406.05590).
> > >
> > > Summary of changes relevant to your comments:
> > >
> > > 1. We have updated the related work section (Section 1.2) as mentioned in our rebuttal
> > > 2. We have fixed the minor edits suggested by you
> > > 3. We have added a case study comparing human and LLM approaches for a CTF challenge in Appendix C

---

> > > > ### Comment · Reviewer_rkZ5 · 2024-08-26
> > > >
> > > > Thank you for providing the revised version. I have updated my review and now recommend acceptance.

---

### Author Response · Authors · 2024-08-19
**Can we send a revision of the paper to the reviewer**

Dear Area Chairs,

We have revised our paper based on the requests from reviewers, and we are wondering if we can share the revised paper with the reviewer via Arxiv or Google Drive? Thanks!

Best regards,

---

### Decision · Program_Chairs · 2024-09-26

**Decision:**

Accept (Poster)

**Comment:**

This paper presents a new dataset for employing LLMs to solve CTF challenges in cybersecurity. It makes the following contributions. First, it provides a scalable dataset for LLM testing, covering different CTF challenges. Second, it builds a fully automated pipeline for comprehensive evaluations of LLM performance in addressing CTF challenges. Third, comprehensive experiments were conducted to evaluate the performance of mainstream LLMs.

The reviewers appreciate the values of the dataset and automated framework. The dataset is well defined and organized. Based on the reviews, this paper is recommended for acceptance. The authors are recommended to continue maintaining this dataset, like incorporating more categories and instances of CTF challenges, and testing more LLMs.